# The Differential Effect of Senolytics on SASP Cytokine Secretion and Regulation of EMT by CAFs

**DOI:** 10.3390/ijms25074031

**Published:** 2024-04-04

**Authors:** Daria A. Bogdanova, Ekaterina D. Kolosova, Tamara V. Pukhalskaia, Ksenia A. Levchuk, Oleg N. Demidov, Ekaterina V. Belotserkovskaya

**Affiliations:** 1Division of Immunobiology and Biomedicine, Sirius University of Science and Technology, Sirius, Krasndarsky Krai, 354340 Sochi, Russia; 2Institute of Cytology RAS, 194064 St. Petersburg, Russia; 3World-Class Research Centre for Personalized Medicine, Almazov National Medical Research Centre, 197341 St. Petersburg, Russia; 4INSERM UMR1231, University of Burgundy, 21078 Dijon, France

**Keywords:** cancer-associated fibroblasts (CAFs), senescence, senescence-associated secretory phenotype (SASP), colorectal cancer, epithelial–mesenchymal transition (EMT), interleukin-6 (IL-6)

## Abstract

The tumor microenvironment (TME) plays an essential role in tumor progression and in modulating tumor response to anticancer therapy. Cellular senescence leads to a switch in the cell secretome, characterized by the senescence-associated secretory phenotype (SASP), which may regulate tumorigenesis. Senolytic therapy is considered a novel anticancer strategy that eliminates the deleterious effects of senescent cells in the TME. Here, we show that two different types of senolytic drugs, despite efficiently depleting senescent cells, have opposite effects on cancer-associated fibroblasts (CAFs) and their ability to regulate epithelial–mesenchymal transition (EMT). We found that senolytic drugs, navitoclax and the combination of dasatinib/quercetin, reduced the number of spontaneously senescent and TNF-induced senescent CAFs. Despite the depletion of senescent cells, the combination of dasatinib/quercetin versus navitoclax increased the secretion of the SASP pro-inflammatory cytokine IL-6. This differential effect correlated with the promotion of enhanced migration and EMT in MC38 colorectal cancer cells. Our results demonstrate that some senolytics may have side effects unrelated to their senolytic activity and may promote tumorigenesis. We argue for more careful and extensive studies of the effects of senolytics on various aspects of tumor progression and tumor resistance to therapy before the senolytic strategy is implemented in the clinic.

## 1. Introduction

Recently, a growing body of evidence suggests that the tumor microenvironment (TME) plays a pivotal role in cancer pathogenesis, including cancer response to antitumor therapy [1]. Among various components of the TME, cancer-associated fibroblasts (CAFs) are considered to be one of the most intriguing entities [2,3], contributing to the malignant phenotype of neoplasms as well as drug resistance, immunosuppression, and increased risk of metastasis [3]. Among the various aspects of CAF biology, the influence of spontaneous or therapy-induced CAF senescence on tumor progression is of particular interest. The senescence process in the TME determines cell fate and disease outcome [4]. Several studies have shown that senescent fibroblasts are abundant and heterogeneous in the TME and are closely associated with cancer progression [5,6]. 

One of the important properties of senescent CAFs is considered to be the senescence-associated secretory phenotype (SASP), which is characterized by the secretion of various pro-inflammatory cytokines, chemokines, growth factors, and metalloproteinases that remodel the extracellular matrix [7]. Nowadays, much attention is paid to the ability of SASP to influence the epithelial–mesenchymal transition (EMT) in cancer cells, which contributes to metastasis and less favorable disease outcome [8]. CAFs can alter the immune landscape of the TME by secreting several factors that suppress immune effector cells, cytotoxic lymphocytes, and NK cells and promote the differentiation of myeloid cells into myeloid-derived suppressive cells. Thus, interactions between CAFs and immune cells in the TME may diminish the antitumor immune response. A better understanding of how CAFs can influence various aspects of tumor growth and help tumors escape immune control may lead to the development of novel cancer treatment strategies. CAFs have been shown to play a critical role in shaping the TME landscape in colorectal cancer (CRC), and targeting them as a therapeutic focus in cancer is a promising approach to improve antitumor therapy [9,10]. Currently, CRC is a significant public health problem and is the second leading cause of cancer-related mortality worldwide [11]. The primary cause of mortality in CRC is metastatic disease. Despite advances in surgical and chemotherapeutic treatment of metastatic CRC, 30% of patients experience metastatic recurrence after initial treatment, and the majority of patients remain incurable with a median survival of less than five years [1]. These observations underscore the need to identify novel adjuvant therapeutic strategies and explore new targets to improve therapeutic approaches for CRC.

In this context, it was hypothesized that pharmacological strategies aimed at eliminating senescent cells may affect inflammation in tumors and reduce the negative effects of SASP, including EMT. This hypothesis has sparked interest in the elimination of senescent cancer cells using so-called “enolytics”—agents that kill senescent cells [12], which can potentially increase the efficacy of cancer treatment and reduce potentially harmful side effects. Since 2016, new chemical compounds capable of selectively targeting and eliminating senescent cells have been reported [13,14,15,16]. Among the senolytics, the most promising drugs include navitoclax and the combination of dasatinib/quercetin [17,18]. The combination of dasatinib/quercetin has been shown to reduce the abundance of senescent cells in vivo without disrupting normal healthy cells [19,20]. Another potent senolytic is navitoclax, a BH3 mimetic that inhibits the anti-apoptotic proteins BCL-2, BCL-W, and BCL-xL [13,21]. This drug selectively eliminates senescent cells in a cell culture by inducing apoptosis. 

In our study, we evaluated the effect of the most promising clinical senolytics, such as navitoclax and dasatinib/quercetin, on the ability to eliminate senescent CAFs in vitro using a unique reporter system for the detection of senescent cells derived from p16-Cre/R26-mTmG mice [22]. In addition, we investigated the effect of removing senescent CAFs from the TME on the migratory activity of murine colorectal cancer cells using the MC38 cell line.

## 2. Results

### 2.1. Expression of Senescence and CAF-Associated Markers Upregulated in Mouse Fibroblasts Following Exposure to Tumor-Cell-Conditioned Medium

Tumor cells can shape their TME and orchestrate severe changes in the phenotype of surrounding cells. Senescent fibroblasts have been shown to exhibit CAF properties [23,24]. The genetic design of the construct in fibroblasts from p16-Cre/R26-mTmG mice allows us to track the appearance of p16^High^ cells in culture. When p16 is expressed, a Cre recombinase is co-expressed. The Cre recombinase excises the stop cassette between two loxP sites from the Rosa 26 locus. Together with the stop cassette, the CRE recombinase removes the gene encoding the ubiquitously expressed red fluorescent protein tdTomato. The excision of the stop cassette makes cells “green” by allowing the expression of EGFP green fluorescent protein. Thus, EGFP-positive cells correspond to p16^High^ senescent cells. (Figure 1A). Figure 1B shows a confocal microscope image illustrating the presence of “green” senescent cells and “red” non-senescent cells among the CAFs. In our experimental design, culturing p16-Cre/R26-mTmG fibroblasts in the conditioned medium of a murine colon adenocarcinoma cell line MC38, which is commonly used to model colon cancer in syngeneic immunocompetent mice, significantly increased the number of p16^High^ senescent cells (Figure 1C).

As expected, key CAF markers, including smooth muscle alpha-actin (*αSMA*) and fibroblast activation protein (*Fap*), were upregulated after 1 day of fibroblast culturing in the conditioned medium of MC38 (CM-MC38) derived from tumor cells (Figure 1D,E). In addition to the expression of CAF markers, CM-MC38-treated mouse fibroblasts showed a significant increase in the expression of the senescence markers *p21* and *p16* (Figure 1F,G). Taken together, these data suggest that fibroblasts acquire characteristics of both CAFs and senescent cells during CAF differentiation.

### 2.2. TNF Signaling Induces Senescence in Murine CAFs 

TNF is one of the major cytokines produced in the TME and is capable of profoundly altering the immune landscape of the TME [25]. TNF exposure has previously been reported to induce premature senescence of endothelial cells [26] and endothelial cell precursors [27], embryonic lung fibroblasts [28]. Moreover, anticancer immunotherapies, such as immune checkpoint blockade (ICB), elevate TNF levels in the TME, and ICB efficiency depends on cytokine-induced senescence in cancer cells [29]. These observations suggest the involvement of TNF in the induction of senescence, but this was mainly studied in cell types other than CAFs in the TME. Our current experimental setup allowed us to trace senescent cells in the CAF population after TNF treatment. 

An increased number of p16^High^ senescent cells was observed after incubation of CAFs with TNF at a concentration of 10 ng/mL. Live-cell imaging shows the onset of the increase in senescent CAFs after 5 days of TNF treatment (Figure 2A). In addition, upregulation of SASP cytokines such as IL1a, IL1b, IL10, and IL-6 was observed (Figure 2B). The upregulation of IL-6 protein, the most prominent cytokine of SASP, was also confirmed at the protein secretion level (Figure 2C). The acquisition of the senescent phenotype was also verified by staining for senescence-associated beta-galactosidase (SA-β-gal) (Figure 2D).

### 2.3. Senolytic Therapy Effectively Removes Senescent CAFs

CAF differentiation with CM-MC38 and TNF signaling both increased the number of senescent fibroblasts. Next, we decided to test the effects of depleting senescent CAFs by senolytic treatment in culture. It is important to note that many types of antitumor therapies, including immunotherapy, lead to an increase in the number of senescent cells in the TME. This has a negative impact on tumorigenesis and resistance to antitumor treatment [30]. Therefore, senolytic therapy is proposed as an adjuvant treatment for oncological diseases to mitigate the negative effects of SASP. 

The most widely used chemical compounds in aging studies for the senolysis assay, navitoclax (N) and the combination of dasatinib/quercetin (D + Q), were selected because of their high clinical potential [4,18]. Using the senescent fibroblast reporter model, we demonstrated a decrease in the number of p16^High^ cells after 5 days of senolytic treatment with navitoclax or the combination of dasatinib/quercetin in both untreated and TNF-treated CAFs (Figure 3A). Furthermore, we observed a downregulation of the expression level of senescence markers such as *p16* and *p21* in cells treated with senolytics (Figure 3B,C).

### 2.4. Differential Effects of Navitoclax and D + Q on SASP-Related IL-6 Secretion

Despite the fact that both senolytic treatments, N and D + Q, efficiently reduced the number of senescent CAFs, their effects on SASP differed, particularly on IL-6 levels. As expected, Navitoclax reduced the number of senescent cells and downregulated IL-6 mRNA expression and the secretion of this cytokine into the surrounding media (Figure 4A,B). Unexpectedly, the D + Q combination showed the opposite effect of increasing IL-6 levels (Figure 4A,B).

In untreated CAFs, we observed a 1.8-fold (*p* < 0.0001) decrease in IL-6 levels relative to control CAFs after navitoclax treatment (Figure 4B). When TNF induced senescence, navitoclax reduced IL-6 levels at both the expression and secretion levels by 5.8-fold and 1.4-fold (*p* < 0.0001, *p* < 0.001, respectively) (Figure 4A,B). In contrast, the combination of dasatinib/quercetin increased IL-6 expression and secretion by 2.1-fold and 1.4-fold, respectively (Figure 4A,B). In TNF-treated CAFs with already elevated IL-6 levels, dasatinib/quercetin potentiated the effect with an additional 1.5-fold increase in IL-6 expression and 1.13-fold increase in IL-6 secretion. This triple drug combination resulted in an 8.7-fold increase in IL-6 expression levels and a 1.6-fold increase in IL-6 secretion levels compared to control CAFs (Figure 4A,B). Finally, we observed a significant increase in IL-6 levels in the conditioned media (CM-CAF) of dasatinib/quercetin-treated CAFs in the TNF-induced senescence model.

### 2.5. The Differences in Senolytic Effects on CAFs’ SASP Determine Epithelial–Mesenchymal Transition in Tumor Cells

The involvement of IL-6 in the regulation of epithelial–mesenchymal transition (EMT) has been highlighted in many studies. We decided to evaluate how conditioned media (CM-CAF) from CAFs treated with different senolytic combinations could affect EMT in the MC38 murine colorectal cancer cell line.

We showed that CM-CAF from fibroblasts treated with navitoclax did not promote the migratory activity of MC38 colorectal cancer cells in the scratch wound healing assay (Figure 5A). In contrast, CM-CAF from D + Q + TNF-treated fibroblasts significantly enhanced the migration of MC38 cells. The observed effect increased the wound healing rate and the migration speed of tumor cells (Figure 5A,B). Notably, this CM-CAF contained the highest levels of IL-6. In addition, the CM-CAF from D + Q + TNF-treated fibroblasts promoted increased expression levels of EMT markers. We observed a strong increase in the expression of the transcription factor Snail in cancer MC38 cells, which is responsible for the transition to a mesenchymal state (Figure 5C). Importantly, the use of navitoclax counteracted the effects of senescent cells, significantly reducing the expression level of Snail (Figure 5C). In addition, the expression of another transcription factor involved in EMT, Zeb1, was significantly decreased only in the presence of navitoclax (Figure 5D).

Suspecting that IL-6, which is an abundant cytokine in the CM-CAF of CAFs treated with D + Q + TNF, might be responsible for EMT in MC38, we used an anti-mouse-IL-6 antibody (IL6inh) to investigate whether it could neutralize these effects on wound healing rate and gene expression involved in EMT. Indeed, the addition of the inactivating mouse-IL-6 antibody to CM-CAF D + Q alleviated the stimulatory effects on the migratory activity of tumor cells. The scratch wound healing assay showed a reduced wound healing rate (Figure 5E,F). Furthermore, in MC38, after the application of antibodies against mIL-6 in CM-CAFs treated with D + Q + TNF, the expression of Snail, but not Zeb1, was dramatically reduced to the level of MC 38 cultured in normal DMEM/FBS (Figure 5C). Although the expression level of Zeb1 was not significantly reduced, there was a trend toward a reduction in Zeb1 expression compared to CM-treated cells from CAFs treated with D + Q + TNF (Figure 5D). The data suggest that IL-6 produced by CAFs in the tumor microenvironment after D + Q treatment plays a critical role and may be responsible for the accelerated migration and EMT signature of MC38 colorectal cancer cells.

Taken together, our results suggest that the combination of dasatinib/quercetin, although it is an effective senolytic treatment, has an undesirable side effect of promoting IL-6-dependent EMT in tumor cells. The observed phenomenon highlights the need for a more detailed study of the influence of these compounds on different cell types in the TME.

## 3. Discussion

The control of cellular aging and SASP control in tumors is considered as a novel treatment strategy for cancer that could potentially be used as an adjunct in antitumor therapy. In recent years, aging fibroblasts/CAFs, which may contribute to tumor progression, evade immune response, and become resistant to antitumor therapy, have emerged as a new target in cancer treatment [31,32]. Novel strategies for targeted elimination of senescent CAFs are under intense scrutiny. Recent successes in this area were published by Nicolas AM and colleagues in Cancer Cell [33], reporting that venetoclax therapy in in vivo CRC models contributes to a reduction in tumor burden. The authors attribute this action to the senolytic effect on senescent CAFs, which, as part of SASP, produce IL-1α, a key factor contributing to CRC resistance to radiotherapy [33]. It is worth noting that venetoclax, the “closest relative” to navitoclax, is also proposed as a senolytic. In our study, navitoclax demonstrates superior in vitro results, suggesting its potential for use as an adjunct therapy. Further detailed investigation of its synergistic action with traditional CRC treatment methods using in vivo models is needed. Similar studies have been conducted using vesicles containing navitoclax targeted at FAP as a CAF marker, demonstrating effectiveness in CAF removal [34]. However, the authors did not explore the combined action with chemotherapeutic agents but emphasized the importance of such research in the future. This could promote the development of a dual strategy with selective efficacy against CAFs and tumor cells, thereby maximizing therapeutic potential and reducing side effects [34]. Another study using a rat model of cholangiocarcinoma demonstrated CAF depletion caused by navitoclax, significantly reducing lymphatic vascularization and metastasis to lymph nodes [35]. 

Besides the senolytic effect of navitoclax and the combination of dasatinib/quercetin reported in our study, contrary to expectations, the effective removal of senescent fibroblasts by the combination of dasatinib/quercetin was accompanied by a significant increase in IL-6, a crucial component of the SASP. In contrast, navitoclax effectively reduced the level of IL-6 in TNF-induced senescence, both at the expression and secretion levels (Figure 6).

It is known that IL-6 is a major pro-inflammatory cytokine with a broad spectrum of negative impacts on tumor cells, contributing to tumor malignancy. IL-6 has been recognized as a potent inducer of CRC progression, including the induction of EMT in HCT116 human colorectal cancer cells [36]. Earlier findings indicate that IL-6 levels, a focal point of this investigation, rise with the advancement of CRC. For example, in a murine model of colitis-associated CRC, obstructing IL-6 signal transmission resulted in a reduction in tumor burden [37].

The unexpected effects of the D + Q combination on the IL-6 levels found in our study may be an obstacle to its use as adjuvant therapy in cancer treatment. Moreover, the upregulation of IL-6 in the CAF model was also observed in another study, where treatment with dasatinib upregulated IL-6 by 2.8-fold compared to the control [38]. A similar effect on SASP was described in Du D et al. The authors reported that in cisplatin-induced senescence, senolytic therapy with dasatinib/quercetin showed a less potent effect on SASP secretion than metformin, and the quantification of another pro-inflammatory cytokine IL-1 β showed an upward trend in cell lysates but did not reach statistical significance compared to cisplatin-induced senescence. However, senolytic therapy with dasatinib/quercetin improved other indices related to ovarian function in cisplatin-induced senescence [39]. These and our data emphasize the importance of studying the effects of senolytics in different models of senescence induction. Many reports indicate that D + Q therapy effectively reduces SASP in replicative senescence [17,19]. There is no question that D + Q therapy effectively eliminates senescent cells. But one must take into account the fact that senescent cells are diverse in nature, and depending on the cell type and method of senescence induction, they may exhibit differences in metabolic, transcriptional, and SASP profiles. Thus, for the proper application of senolytics, we require extensive studies aimed at characterizing different types of senescent cells in culture and especially in vivo. Although the therapeutic benefits of senolytics have accelerated their path from discovery to clinical trials (for example, D + Q therapy is involved in a number of clinical trials: chronic kidney disease (NCT02848131), idiopathic pulmonary fibrosis (NCT02874989), skeletal health (NCT04313634), hematopoietic stem cell transplant survivors (NCT02652052), Alzheimer’s disease (NCT04063124), and adult cancer survivors (NCT04733534)), not all senolytics are ideal in their current state. For example, despite proven efficacy in our in vitro system, off-target effects have been observed with navitoclax, which shows evidence of platelet toxicity in the clinic. However, despite the limitations and side effects due to the development of intermittent dosing regimens that are required for senolytics, it is possible to offset the toxic effects [40]. 

Previously published data combined with our results further emphasize the complex nature of senescent cells. Senolytic therapy appears to be an excellent option for regulating the undesirable negative effects associated with senescent cell accumulation both during cancer treatment and during oncogenesis, where paracrine tumor cells contribute to TME senescence. A more detailed study of the effects of different senolytic approaches in appropriate models of senescence in vitro and in vivo is required.

## 4. Materials and Methods

### 4.1. Derivation and Culture of Primary Mouse Dermal Fibroblasts (DFs)

DFs isolated from newborn (not older than 2 days after birth) pups from p16Cre; mTmG mice were used as culture CAF cells. The back skin of the newborn pups was treated with dispase II (#D4693, Sigma-Aldrich, Saint Louis, MO, USA) (2.5 mg ml^−1^ in Dulbecco’s modified Eagle’s medium (DMEM, Gibco BRL, Invitrogen, Grand Island, NY, USA) overnight at +4 °C; then, it was transferred to a collagenase IV solution (Gibco BRL, Invitrogen, Grand Island, NY, USA) (1 mg ml^−1^ in DMEM) for 1 h at room temperature. Collagenase was inactivated through the addition of DMEM with 10% fetal bovine serum (FBS, Capricorn Scientific GmbH, Ebsdorfergrund, Germany), and the cell suspension was successively passed through nylon mesh strainers with pore diameters of 100, 70, and 40 μm. Cells were resuspended in DF culture medium (DMEM, 10% FBS, 1× penicillin–streptomycin (Capricorn Scientific GmbH, Germany)) and plated on a 150 mm Petri dish. This passage was considered as zero, and the cells were frozen after genotyping. Dermal fibroblast cultures were subjected to differentiation into CAFs by incubation with CM-MC38 for 24 h. After that, the CAF culture was treated with senolytics for 5 days with a change of medium with treatment on day 1 and day 3. DMEM/FBS supplemented with 0.5 µM navitoclax (#HY-10087, MedChemExpress, Monmouth Junction, NJ, USA), 0,1 µM, with dasatinib (#SML2589, Sigma-Aldrich, Saint Louis, MO, USA) and 1 µM quercetin (HY-18085, MedChemExpress, USA), or with 10 ng/mL TNF r with 10 ng/mL TNF (from *E. coli* purified as described previously [41]), or with a combination of TNF and N or TNF and D+Q. The control CAFs were cultured with replacement of the regular DMEM/FBS medium on the corresponding days. Fluorescence signal detection occurred in an Incucyte lifetime imaging device every 4 h for the duration of treatment.

### 4.2. Preparation of Conditional Media

MC38 cells (10^6^) were seeded in DMEM (Gibco BRL, Invitrogen, Grand Island, NY, USA) supplemented with 10% FBS (Capricorn Scientific GmbH, Ebsdorfergrund, Germany), 2 mM of L-glutamine (Gibco BRL, Invitrogen, Grand Island, NY, USA), and antibiotics (100 U/mL of penicillin, 100 μg/mL of streptomycin, Capricorn Scientific GmbH, Germany) on a 10 cm^2^ dish and incubated at 37 °C with 5% CO_2_ and allowed to reach confluence for 3 days. The conditioned medium was prepared in a 1:1 ratio with DMEM and 10% FBS, and it was added to the DFs for 24 h. The conditioned medium from CAFs was collected after the completion of treatment by adding fresh DMEM without any drugs and supplemented with 10% FBS (for qPCR) or without FBS (for wound healing assay). This conditioned medium was added to the MC38 culture for 48–72 h to assess the expression of epithelial–mesenchymal transition (EMT) genes or for the wound healing assay. To inhibit IL-6 in CM-CAF, an antibody against murine IL-6 was used in experiments at a final concentration of 0.01 mg/mL (Bio Cell, InVivoMAb anti-mouse IL-6, Lebanon, PA, USA).

### 4.3. RNA Isolation and qPCR

Total RNA was isolated from fibroblasts using Trizol (Invitrogen, Carlsbad, CA, USA) according to the manufacturer’s protocol. Single-stranded complementary DNA (cDNA) was obtained from a reverse transcription of 500 ng of RNA using an RT-PCR kit (SuperScript™ II Reverse Transcriptase, Invitrogen, Carlsbad, CA, USA). qRT-PCR was performed using SYBR Green (5X qPCRmix-HS SYBR + LowROX, Evrogene, Moscow, Russia) according to the manufacturer’s instructions. Data were normalized to *Gapdh* expression. Each sample was analyzed in triplicate. Gene expression was analyzed using the 2^−ΔΔCt^ method. Primer sequences are listed in Appendix A.

### 4.4. Enzyme-Linked Immunosorbent Assay (ELISA)

The IL-6 cytokine in cell supernatants was estimated by ELISA, using a commercial kit (ELISA MAX™ Deluxe Set Mouse IL-6, BioLegend, San Diego, CA, USA), according to the manufacturer’s instructions. Positive controls were supplied in the kit.

### 4.5. Wound Healing Assay

MC38 cells were planted on 96-well plates (4 × 10^4^ cells/well). To assess the migratory activity of MC38 cells in the scratch assay, cells were incubated in a serum-free medium overnight. Subsequently, a scratch was made using a specialized Wound Maker device (IncuCyte), and the medium was replaced with serum-free CM-CAF. The wound healing was then detected using the IncuCyte live-cell imaging system every 4 h for 72 h.

### 4.6. Statistical Analysis

Comparisons between two groups were analyzed with the Student’s t-test, and differences among multiple groups with one-way ANOVA with Tukey’s multiple comparisons test using GraphPad Prism 5.0 (GraphPad Inc., La Jolla, CA, USA). Values of *p* < 0.05 were considered statistically significant. Data were represented as mean ± SEM.

## 5. Conclusions

Our data in an in vitro mouse colorectal cancer model on the MC38 cell line show the dasatinib/quercetin combination to successfully eliminate senescent CAFs alongside a significant elevation in secretory IL-6 levels. Moreover, the increment in IL-6 levels subsequently enhanced the migratory potential and EMT of CRC tumor cells. In contrast, navitoclax not only effectively eliminated senescent CAFs but also reduced IL-6 levels in TNF-induced senescence (Figure 6).

These results emphasize the importance of accounting for the effects of the senescence-associated secretory phenotype (SASP), which may contribute to cancer development. Future clinical trials comparing senolytic protocols to standard CRC treatments are critical to determine their efficacy and potential side effects. Studies investigating the interaction between tumor microenvironment aging and CRC hold promise to optimize treatment strategies and improve overall patient survival.

## Figures and Tables

**Figure 1 ijms-25-04031-f001:**
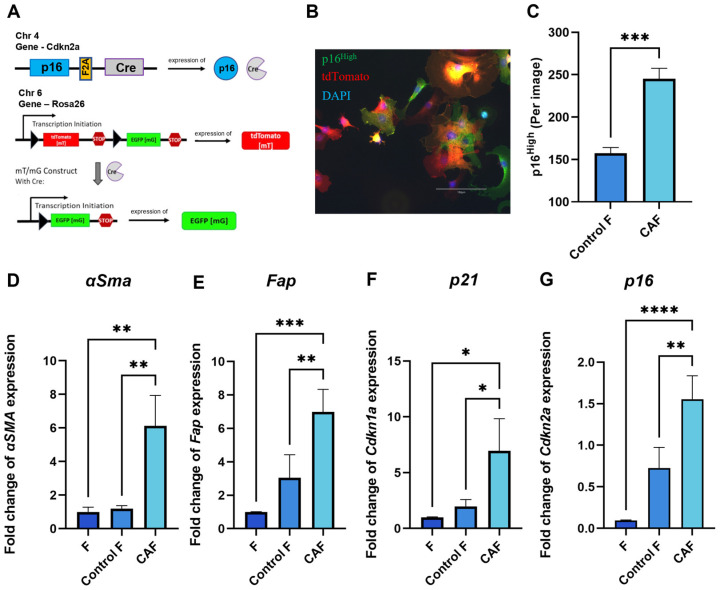
Acquisition of CAF properties in association with fibroblast senescence. (**A**) Schematic of the mouse p16-Cre/R26-mTmG genetic construct. Cre recombinase expression is linked to the expression of p16, one of the major markers of senescent cells. In p16^High^ senescent cells, Cre recombinase silences tdTomato red fluorescent protein, thus allowing the expression of green fluorescent protein EGFP, which labels senescent cells. (**B**) Representative image of a p16-Cre/R26-mTmG murine fibroblast culture, scale bar 150 μm (**C**) The number of p16^High^ cells per image shown with the IncuCyte lifetime imaging device. Data were analyzed using two-tailed Student’s t-test; *** *p* < 0.001. The expression of (**D**) *Fap*, (**E**) *α-Sma*, (**F**) *Cdkn1a* (*p21*), (**G**) *Cdkn2a* (*p16*) in fibroblasts was analyzed by qPCR. Charts represent relative expression levels as mean values ± SEM normalized by *Gapdh*. Data from one out of three independent experiments are shown. Data were analyzed using one-way ANOVA with Tukey’s multiple comparisons test; * *p* < 0.05, ** *p* < 0.01, *** *p* < 0.001, **** *p* < 0.0001. F—“young” fibroblasts on 2nd passage; Control F—fibroblasts incubated in DMEM for 1 day; CAF—cancer-associated fibroblast incubated in CM-MC38 for 1 day.

**Figure 2 ijms-25-04031-f002:**
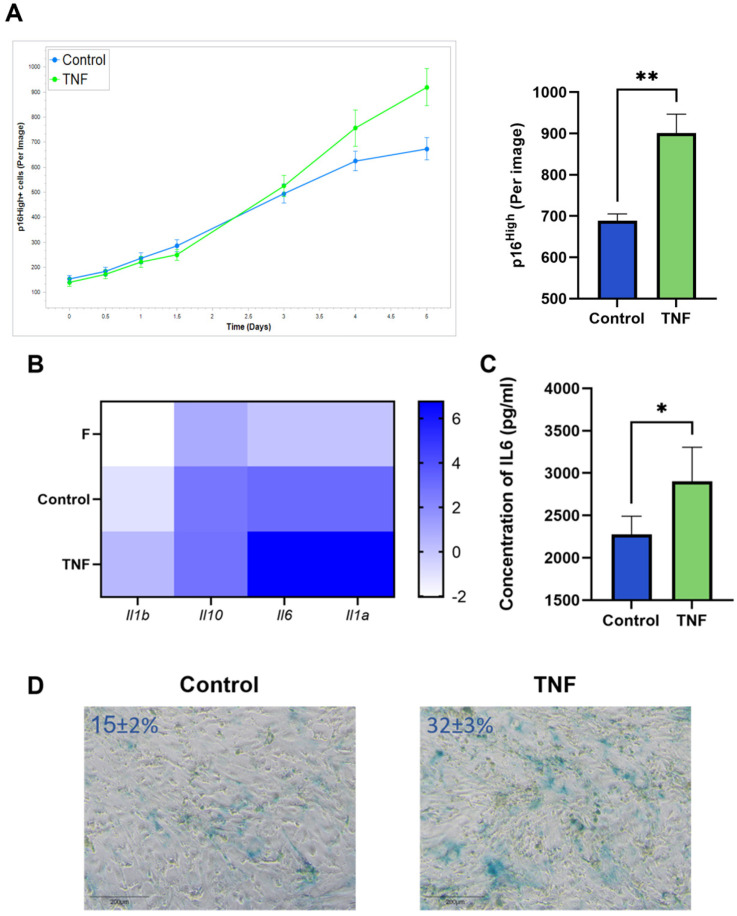
TNF induces senescence in CAFs. (**A**) The number of green p16^High^ senescent cells per image shown with the IncuCyte lifetime imaging system as a time-dependent curve (top left) and histogram (top right). (**B**) Heatmap of the cytokine gene expression, major components of SASP. (**C**) Concentration of IL-6 in supernatants was measured by ELISA. (**D**) Representative image of SA-β-gal staining CAFs (SA-β-gal + cells ± s.d.%). In all images scale bar 200 μm. Data are shown as mean values ± SEM (*n* = 3). Data were analyzed using two-tailed Student’s t-test; * *p* < 0.5, ** *p* < 0.01. F—“young” fibroblasts; Control—CAFs incubated in regular DMEM/FBS medium for 5 days; TNF—CAFs incubated in DMEM/FBS supplemented with 10 ng/mL TNF for 5 days.

**Figure 3 ijms-25-04031-f003:**
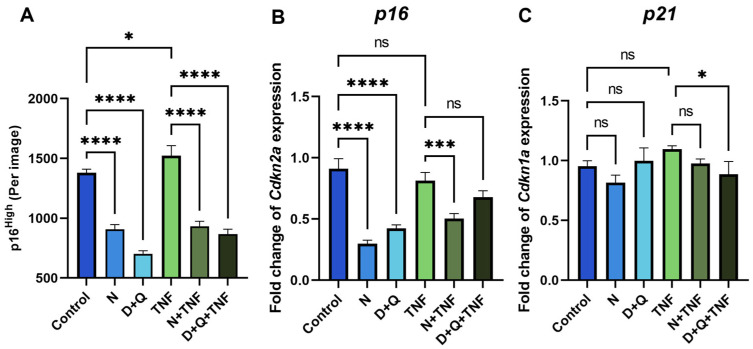
Navitoclax and the combination of dasatinib/quercetin effectively remove senescent CAFs. (**A**) The number of green EGFP-po reporter CAFs per image shown with the IncuCyte lifetime imaging device. The expression of (**B**) *Cdkn1a* (*p16*) and (**C**) *Cdkn1a* (*p21*) in CAFs was analyzed by qPCR. Charts represent relative expression levels as mean values ± SEM normalized by *Gapdh*. Data from one out of three independent experiments are shown. Data were analyzed using one-way ANOVA with Tukey’s multiple comparisons test; * *p* < 0.05, *** *p* < 0.001, **** *p* < 0.0001, ns—not significant. Control—CAFs incubated in DMEM/FBS for 5 days where replicative senescence gradually increased; N—CAFs incubated in DMEM/FBS supplemented with 0.5 µM navitoclax for 5 days; D + Q—CAFs incubated in DMEM/FBS supplemented with 0.1 µM dasatinib and 1 µM quercetin for 5 days; TNF—CAFs incubated in DMEM/FBS supplemented with 10 ng/mL TNF for 5 days.

**Figure 4 ijms-25-04031-f004:**
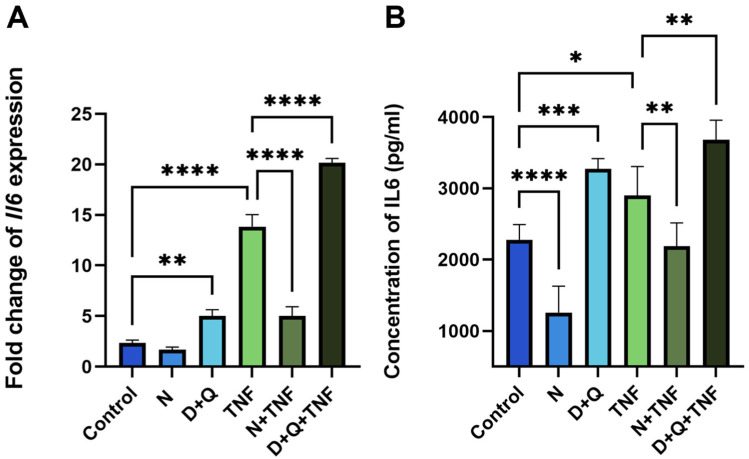
Navitoclax and the combination of dasatinib/quercetin differentially regulate IL-6 expression and secretion levels. (**A**) The expression of IL-6 in CAF cells was evaluated using qPCR. Charts represent relative expression levels as mean values ± SEM normalized by *Gapdh*. (**B**) The IL-6 concentration in supernatants was measured by ELISA. Data are presented as mean ± SD (*n* = 3 replicates per group); data were analyzed using one-way ANOVA with Tukey’s multiple comparisons test; * *p* < 0.05, ** *p* < 0.01, *** *p* < 0.001, **** *p* < 0.0001. Control, N, D+Q - different shades of blue, TNF, N+TNF, D+Q+TNF - different shades of green. Control—CAFs incubated in DMEM/FBS for 5 days where replicative senescence gradually increased; N—CAFs incubated in DMEM/FBS supplemented with 0.5 µM navitoclax for 5 days; D + Q—CAFs incubated in DMEM/FBS supplemented with 0.1 µM dasatinib and 1 µM quercetin for 5 days; TNF—CAFs incubated in DMEM/FBS supplemented with 10 ng/mL TNF for 5 days.

**Figure 5 ijms-25-04031-f005:**
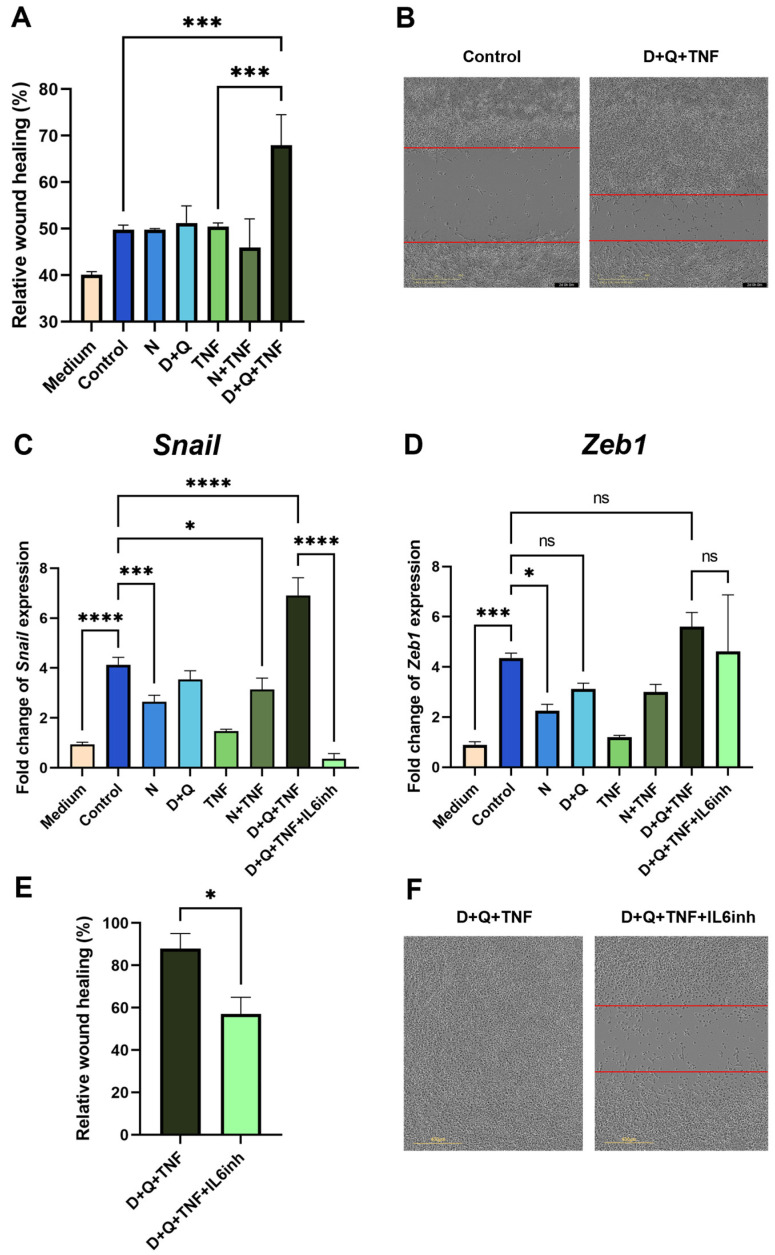
CAFs treated with the D + Q combination promoted the expression of EMT markers and enhanced the migration of MC38 cancer cells. (**A**) Analysis of cell migration activity in the scratch wound healing assay quantified using the IncuCyte live-cell imaging system. Mean migration/invasion values ± SEM (*n* = 3). (**B**) Representative images of MC38 cells in the scratch wound healing assay at 0 and 48 h in areas with the addition of CM from untreated CAFs and CAFs treated with D + Q + TNF. (**C**) Expression of Snail and (**D**) Zeb1 in CAFs was analyzed by qPCR. Charts represent relative expression levels as mean values ± SEM normalized by Gapdh. Data from one of three independent experiments are shown. (**E**) Analysis of cell migration activity in the scratch wound healing assay. Mean migration/invasion values ± SEM (*n* = 3). (**F**) Representative images of MC38 cells in the scratch wound healing assay at 72 h in areas with the addition of CM from CAFs treated with D + Q + TNF with or without anti-IL-6 antibodies (IL-6inh). In all images scale bar 400 μm. Data were analyzed using one-way ANOVA with Tukey’s multiple comparisons test; * *p* < 0.05, *** *p* < 0.001, **** *p* < 0.0001, ns—not significant.

**Figure 6 ijms-25-04031-f006:**
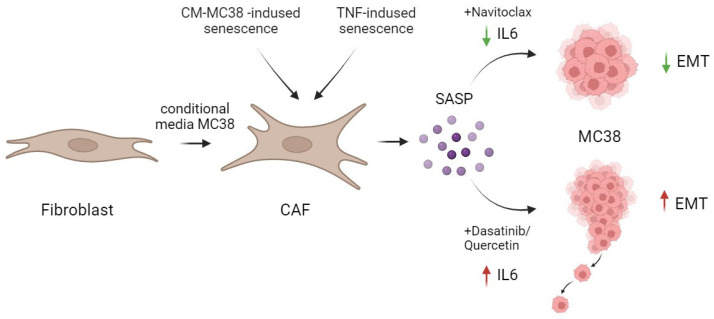
Schematic representation of the action of senolytics in the treatment of CAFs. The combination of dasatinib/quercetin versus navitoclax increased the secretion of one of the major pro-inflammatory cytokines in SASP—IL-6. This differential effect correlated with the promotion of enhanced migration and EMT in MC38 colorectal cancer cells. Created by Biorender. by Biorender https://www.biorender.com/ (accessed on 1 December 2023).

## Data Availability

The data that support the findings of this study are available from the corresponding authors.

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
