# Peer review of "The Differential Effect of Senolytics on SASP Cytokine Secretion and Regulation of EMT by CAFs"

_ijms, 2024, doi:10.3390/ijms25074031_

Round 1

Reviewer 1 Report

Comments and Suggestions for Authors

The article by Bogdanova et al. discusses an important topic: whether senolytic therapeutics can be used to eliminate cancer-associated fibroblasts (CAFs) to potentially prevent tumor progression. The authors employed a system to induce the CAF phenotype in mouse fibroblasts by treating them with conditioned media from the CRC cell line MC38 or the inflammatory chemokine TNF. They demonstrated that both treatments induced a senescence-associated secretory phenotype (SASP) in fibroblasts. Two different types of senolytic drugs, navitoclax and the combination of dasatinib/quercetin, reduced the number of senescent CAFs, but had opposite effects on the production of IL6. Dasatinib/quercetin increased the secretion of IL-6, thereby enhancing epithelial-mesenchymal transition (EMT) in tumor cells, warranting a careful evaluation of unwanted side effects before implementation of synolytics as anticancer drugs. While the manuscript is well-written and supported by good-quality data, some minor revisions are recommended to improve it before publication.
1. The title does not fully represent the manuscript, lacking the term CAF which might give the impression that SASP is studied in tumor cells rather than CAFs. I would recommend improving the title.
2. While myCAF markers (aSMA, FAP) were tested in fibroblasts in response to conditioned media (CM), inflammatory markers were not. However, it is known that conditioned media from tumor cells stimulates a more inflammatory CAF phenotype in fibroblasts, and TNF for sure induces inflammatory CAFs. Please discuss the impact of CM on the polarization of fibroblasts or demonstrate it by including a few inflammatory genes into the list.
3. The induction of inflammatory CAF genes is relatively quick in fibroblasts, within 24 hrs, but the senescent phenotype was observed 5 days after treatment with CM or TNF. Do you think that immediate induction of inflammation in CAFs can lead to senescence, but not the other way around?
4. Navitoclax is an inhibitor of BCL2. Do you observe the induction of BCL2 in CAFs? If not, what does navitoclax target?
5. Please discuss which cells produce TNF. I am not sure whether tumor cells are the main source of TNF in the TME.
6. Please describe in the legends what F and control (Fig. 2) represent.
7. In Fig 2D, did you measure the level of senescence in young fibroblasts?
8. Overall, in the manuscript, please indicate what is on the graph: biological replicates, technical replicates, n=?.
9. Unify anti-tumor and antitumor (no dash).
10. In Fig 1D-G, please add units on the Y-axis.
11. Gene names should be italicized (e.g., Cdkn2).
12. In the Materials and Methods section, please add a reference for p16cre mice or a depository number in JAX or elsewhere.
13. I assume that navitoclax might be actively used for leukemia treatment (or mostly for CLL). Are there any studies showing that navitoclax can prevent relapse of leukemic blasts in the bone marrow niche by eliminating senescent mesenchymal stromal cells?
14. In the graphical abstract, only one cell is presented, but fibroblasts is written in the plural form. Navitoclax is misspelled as nanitoclax.
15. The title 2.1. Exposure to tumor cell conditioned medium leads to upregulated expression of senescence and CAF-associated markers in mouse fibroblast" appears to be grammatically incorrect.

Author Response

Thank you very much for taking the time to review this manuscript! Please find the detailed responses below and the corresponding corrections highlighted yellow changes in the re-submitted files.

Reviewer 2 Report

Comments and Suggestions for Authors

The manuscript is very interesting and the study is well-conducted. 

There are, however, some points which the Authors need to address:

1) Introduction - please provide more insights on how tumour/cancer-associated fibroblasts contribute to development of immunosuppressive TME. And explain the aim of your work in light of this. 

2) Results are well presented. There are several minor issues here:

a) Figure 1 B please provide a more detailed description of the results depicted in this image both in the figure legend and in the manuscript text, otherwise it takes time to understand. 

b) Figure 2. And other Figures. Which isoform of TNF was studied? Please specify.

c) There are obvious errors in statistical calculations: 

Figure 2A (Bar diagram) and 2 C (bar diagram) - in the first case it is ** and not **, in the second it is * at best, but not **. 

Figure 3 - differences between Control and TNF are * at best. 

Figure 4 B - Differences between TNF and N+TNF are * at best, but definitely not ***

Figure 5 E - differences can't be significant with such a high SEMs 

4) Discussion. Needs to be more conceptual. Please provide an indication of how your findings contribute to understanding the insights of the immunosuppressive TME. How do cytokines you studied contribute to it ( e g - activation of immune evasion machinery). 

5) Figure 6. It says graphical abstract. Graphical abstract needs to be either at the beginning of the manuscript or just call the figure - "Scheme representing... Also please expand this scheme and provide the insights of what does it mean in terms of biological response/tumour progression. Otherwise, by looking at this scheme, one just has a question: "so what?"

Comments on the Quality of English Language

I would suggest that a native speaker reads the paper through. Overall language is OK, but some of the sentences are really large and thus difficult to follow (one needs to read these sentences several times in order to understand the Authors' meaning). Just an example here -first sentence in the Conclusions: 

"Our data in an in vitro mouse colorectal cancer model on MC38 cell culture show that the combination of dasatinib and quercetin, although successfully eliminated senescent CAFs, significantly increased IL-6 levels, which subsequently enhanced the migratory potential and EMT of CRC tumor cells" This is a very important statement, but really hard to follow. It needs to be broken up into two simple sentences.

And such issues appear on a number of occasions in the text. So please check and edit. 

Author Response

Thank you very much for taking the time to review this manuscript! Please find the detailed responses below and the corresponding corrections highlighted yellow changes in the re-submitted files.

In the new version of the manuscript, we have also paid more attention to the complex sentences in the English language. We have tried to make the text easier to understand by breaking it down into simpler components.

Round 2

Reviewer 2 Report

Comments and Suggestions for Authors

The Authors have addressed my concerns. 

Just one thing - in the literature we still use abbreviation TNF-a and not just TNF to indicate this specific cytokine, which is called tumour necrosis factor alpha. 

I leave this with the Authors, since this is not an error. The rest of the comments were satisfactory addressed. 

Comments on the Quality of English Language

See sbove